# Hexagonal and Monoclinic Phases of La_2_O_2_CO_3_ Nanoparticles and Their Phase-Related CO_2_ Behavior

**DOI:** 10.3390/nano10102061

**Published:** 2020-10-19

**Authors:** Hongyan Yu, Kaiming Jiang, Sung Gu Kang, Yong Men, Eun Woo Shin

**Affiliations:** 1School of Chemical Engineering, University of Ulsan, Daehakro 93, Nam-gu, Ulsan 44610, Korea; yuyubobo0908@163.com (H.Y.); hg041111138@163.com (K.J.); sgkang@ulsan.ac.kr (S.G.K.); 2College of Chemistry and Chemical Engineering, Shanghai University of Engineering Science, Shanghai 201620, China; men@sues.edu.cn

**Keywords:** monoclinic, hexagonal, La_2_O_2_O_3_ phase, CO_2_ behavior, precipitation method, hydrothermal method

## Abstract

In this study, we prepared hexagonal and monoclinic phases of La_2_O_2_CO_3_ nanoparticles by different wet preparation methods and investigated their phase-related CO_2_ behavior through field-emission scanning microscopy, high-resolution transmission electron microscopy, Fourier transform infrared, thermogravimetric analysis, CO_2_-temperature programmed desorption, and linear sweeping voltammetry of CO_2_ electrochemical reduction. The monoclinic La_2_O_2_CO_3_ phase was synthesized by a conventional precipitation method via La(OH)CO_3_ when the precipitation time was longer than 12 h. In contrast, the hydrothermal method produced only the hexagonal La_2_O_2_CO_3_ phase, irrespective of the hydrothermal reaction time. The La(OH)_3_ phase was determined to be the initial phase in both preparation methods. During the precipitation, the La(OH)_3_ phase was transformed into La(OH)CO_3_ owing to the continuous supply of CO_2_ from air whereas the hydrothermal method of a closed system crystallized only the La(OH)_3_ phase. Based on the CO_2_-temperature programmed desorption and thermogravimetric analysis, the hexagonal La_2_O_2_CO_3_ nanoparticles (HL-12h) showed a higher surface CO_2_ adsorption and thermal stability than those of the monoclinic La_2_O_2_CO_3_ (PL-12h). The crystalline structures of both La_2_O_2_CO_3_ phases predicted by the density functional theory calculation explained the difference in the CO_2_ behavior on each phase. Consequently, HL-12h showed a higher current density and a more positive onset potential than PL-12h in CO_2_ electrochemical reduction.

## 1. Introduction

Recently, the synthesis of nanomaterials with controllable morphologies and phases has attracted considerable attention in the fields of materials science and inorganic chemistry because the physicochemical and structural properties of the nanomaterials strongly correlate with the types of crystal structures as well as the morphologies of nanoparticles [1,2,3,4,5,6]. The unique properties of nanomaterials can be properly tuned by controlling various factors, which results in potential applications of nanomaterials in catalysis, biological labeling, sensing, and optics [1,7,8,9]. Among the methods for synthesizing nanomaterials, wet chemical processes have been considered as the most effective and convenient approaches for the controllable phases of ceramic materials [10].

Lanthana (La_2_O_3_) has been widely used as a promoter or support in heterogeneous catalysis [11,12,13]. The basicity of La_2_O_3_ readily induces the adsorption of CO_2_ to form the lanthanum oxycarbonate (La_2_O_2_CO_3_) phase, which is an important species in the La_2_O_3_-containing catalytic reaction [4,13,14,15,16]. The crystalline structures of La_2_O_2_CO_3_ can be divided into three types of different polymorphs: a tetragonal La_2_O_2_CO_3_ (type I), a monoclinic La_2_O_2_CO_3_ (type Ia), and a hexagonal La_2_O_2_CO_3_ (type II) [16,17,18]. The hexagonal type II La_2_O_2_CO_3_ has a higher chemical stability to water and carbon dioxide than the monoclinic type Ia [4,19,20]. In addition, the different crystalline structures of the La_2_O_2_CO_3_ phases affect the interaction between La_2_O_2_CO_3_ and ZnO in the La_2_O_2_CO_3_/ZnO composite materials as well as the catalytic behavior of the composite materials on glycerol carbonation with CO_2_ [4,21]. Meanwhile, the monoclinic type Ia La_2_O_2_CO_3_ phase closely resembles the crystalline structure of lanthanum (La) oxysalts (e.g., oxysilicates, oxyhalides, and oxysulfates), whereas the hexagonal type II one is similar to A-type La sesquioxides. Thus, the type Ia La_2_O_2_CO_3_ phase has been readily prepared by the thermal decomposition of La compounds (e.g., oxalates and acetates); however, it is difficult to prepare type II La_2_O_2_CO_3_ in a single phase by the conventional wet preparation methods [20]. Accordingly, it is necessary to investigate i) the preparation conditions used to form type Ia and type II La_2_O_2_CO_3_ phases in the conventional methods and ii) the CO_2_ behavior on the La_2_O_2_CO_3_ structures, which is an essential step in the CO_2_-involving catalytic reactions, as well as the formation of the different La_2_O_2_CO_3_ phases.

In this study, we prepared the nanoparticles with type Ia and type II La_2_O_2_CO_3_ crystal structures by conventional wet preparation methods and investigated the formation of different La_2_O_2_CO_3_ phases with Fourier transform infrared (FT-IR), X-ray diffraction (XRD), field-emission scanning electron microscopy (FE-SEM), and high-resolution transmission electron microscopy (HR-TEM). Furthermore, the CO_2_ behavior on the different La_2_O_2_CO_3_ crystal structures was observed by CO_2_-temperature programmed desorption (TPD), thermogravimetric analysis (TGA), and linear sweeping voltammetry (LSV) of CO_2_ electrochemical reduction. The superior CO_2_ behavior of the hexagonal La_2_O_2_CO_3_ phase to the monoclinic phase was additionally explained by the crystalline structures of both La_2_O_2_CO_3_ phases, which was predicted by the density functional theory (DFT) calculation.

## 2. Materials and Methods

### 2.1. Materials

A total of 1.00 g of La(NO_3_)_3_·6H_2_O was added to 50.0 mL of deionized water, and the resultant solution was vigorously stirred to ensure complete dissolution. The pH of the solution was adjusted to 12 with a 10 wt% NaOH solution, which yielded a white precipitate after the mixture was stirred for approximately 10 min. The sample was continuously stirred for another 6, 12, or 24 h, and the obtained product was centrifuged. The separated precipitate was washed with distilled water and ethanol and then dried at 80 °C for 12 h, followed by the calcination step at 500 °C for 2 h. Depending on the precipitation time, the solid samples prepared by the precipitation method were denoted as PL-*x*h (*x* = 6, 12, or 24), where *x* represents the precipitation time.

For the hydrothermal method, the procedure was almost the same as that in the precipitation method, except using an autoclave for the hydrothermal treatment. The pH-adjusted solution containing the La precursor was transferred to an autoclave (200 mL), heated to 160 °C, and maintained at this temperature for 6, 12, or 24 h. The obtained product was centrifuged, and the remained steps were also the same as those in the precipitation method. The La_2_O_2_CO_3_ samples synthesized by the hydrothermal method were designated as HL-*y*h (*y* = 6, 12, or 24), where *y* represents the hydrothermal treatment time.

### 2.2. Characterizations

The morphologies of the samples were observed by a field-emission scanning electron microscope (JEOL, JSM-600F, Tokyo, Japan) instrument equipped with an energy-dispersive spectrometer. HR-TEM images were obtained using a JEOL JEM-2100F instrument (JEOL Ltd., Tokyo, Japan). The samples were prepared by suspending and grinding in an ethanol solution whose drops were placed on a carbon-film-coated copper grid. XRD patterns were measured at room temperature on a Rigaku D/MAX-2200 powder X-ray diffractometer (Rigaku Corporation, Tokyo, Japan) using a Cu Kα radiation source (λ = 0.15418 nm). The X-ray tube was operated at 35 kV and 20 mA, and the 2θ angle was scanned from 10° to 90° (with a step of 0.02°) at a speed of 2°/min. The FT-IR spectra of the samples were collected for the KBr powder-pressed pellets on a Nicolet 380 FT-IR spectrophotometer (Thermo Fisher Scientific, Waltham, MA, USA) under ambient conditions.

The CO_2_-TPD experiments were conducted in a quartz flow reactor. The calcined samples were preheated from room temperature up to 600 °C (with a ramping rate of 15 °C/min) for 1 h under He flow (100 mL/min). The CO_2_ gas (10 vol.% CO_2_/He) was fed into the reactor with a flow rate of 30 mL/min at 50 °C for CO_2_ adsorption before conducting the CO_2_-TPD measurements. Finally, the temperature was increased from 50 to 600 °C at the ramping rate of 1.5 °C/min in He flow (30 mL/min). The weight loss in the samples was determined by a thermogravimetric analyzer (TA Instruments Q50, New Castle, DE, USA). A total of 20 mg of the samples was charged into the sample pan and heated to 1000 °C at a rate of 5 °C/min in air flow. The CO_2_ electrochemical reduction was carried out via the LSV measurement with an Ag/AgCl electrode as a reference electrode and Pt wire as a counter electrode. The working electrode was prepared by dispersing 10 mg of the samples in a mixture of 2 mL of alcohol and 100 μL of 5% Nafion and then pipetting 10 μL of suspension on the GCE (0.07065 cm^2^). The working electrode was tested 20 times at a scan rate of 20 mV/s. The electrolyte was 0.1 M NaHCO_3_ saturated with CO_2_. Before each experiment, high-purity CO_2_ gas was bubbled at a flow rate of 30 mL/min for 30 min to remove all oxygen from the electrolyte. The gases in the measurement were analyzed by a GC instrument.

Using the Vienna Ab initio Simulation Package (VASP) [22,23], DFT calculations were conducted along with the GGA–PBE (Perdew–Burke–Ernzerhof) functional [24]. The cutoff energy of 600 eV was chosen in our calculations. The criteria of convergence of energies and forces for geometry optimization were 10^−4^ eV and 10^−2^ eV/Å, respectively. For the calculation of disordered hexagonal La_2_O_2_CO_3_, the lowest energy configuration among the other randomly selected 50 structures was used. The Monkhorst-Pack *k*-point meshes of 3 × 5 × 2 and 9 × 9 × 3 were used for the geometry optimization of monoclinic and hexagonal phase of La_2_O_2_CO_3_, respectively [25].

## 3. Results and Discussion

### 3.1. Synthesis of Monoclinic and Hexagonal La_2_O_2_CO_3_ Nanoparticles

Figure 1 shows the XRD patterns of La_2_O_2_CO_3_ nanoparticle materials prepared at each reaction time. The two types of La_2_O_2_CO_3_ phases are primarily detected in the PL samples: the monoclinic type Ia and hexagonal type II La_2_O_2_CO_3_ phases. For 6 h of precipitation (PL-6h), the characteristic XRD peaks in the hexagonal La_2_O_2_CO_3_ crystal phase are clearly observed at 2θ = 25.7, 30.2, 47.2, and 56.6° (JCPDS 37-0804) (Figure 1(Aa)) [1,4,20,21,26,27]. However, when the precipitation time is increased to 12 and 24 h, the characteristic XRD peaks in the monoclinic La_2_O_2_CO_3_ phase clearly appear at 2θ = 22.8, 29.3, 31.0, 39.9, and 44.4° with a C12/c1 space group (JCPDS 48-1113) (Figure 1(Ab,Ac)), which indicates the prevalence of the hexagonal La_2_O_2_CO_3_ phase during the initial precipitation time, followed by the transformation into the monoclinic La_2_O_2_CO_3_ phase after 12 h of precipitation. In contrast, the HL samples show the XRD patterns that contain the characteristic peaks in only the hexagonal type II La_2_O_2_CO_3_ phase, regardless of the reaction time during the hydrothermal preparation, which demonstrates that there is no change in the La_2_O_2_CO_3_ phase during the preparation process (Figure 1(Ba–Bc)).

The FT-IR spectra of the PL and HL samples also confirm the formation of each La_2_O_2_CO_3_ crystal phase depending on the preparation methods, as shown in Figure 2. According to the assignments of typical FT-IR bands for carbonates in the La_2_O_2_CO_3_ phases, the bands at 745, 855, 1066, and 1518 cm^−1^ are interpreted as CO_3_^2−^ vibrations related to the La_2_O_2_CO_3_ phase [4,6,21,27]. The three-fold splitting bands at approximately 845 cm^−1^ (υ_2_) and a strong band at 1367 cm^−1^ (υ_3_) are assigned to the unique carbonate vibrational mode for the monoclinic type Ia La_2_O_2_CO_3_ phase. The FT-IR spectra in Figure 2b,c of only the PL-12h and PL-24h samples show the characteristic bands (υ_2_ and υ_3_) of type-Ia La_2_O_2_CO_3_, whereas the FT-IR spectra of the other samples show the typical bands of the La_2_O_2_CO_3_ phase, which further confirms that the formation of the type Ia and II La_2_O_2_CO_3_ phases depends on the preparation conditions. In the precipitation method, the monoclinic type Ia La_2_O_2_CO_3_ phase is mainly formed when the precipitation time is longer than 12 h, whereas the hydrothermal method produces only the hexagonal type II La_2_O_2_CO_3_ phase. This is consistent with the XRD results in this study.

Moreover, TEM measurements also provide additional evidence for the existence of the monoclinic and hexagonal La_2_O_2_CO_3_ phases in the samples. Figure 3a–c shows the TEM images and fast Fourier transform patterns of PL-6h, PL-12h, and HL-12h. The (207) plane of the monoclinic type Ia La_2_O_2_CO_3_ phase is detected in the PL-12h sample, whereas the (260) plane of the hexagonal type II La_2_O_2_CO_3_ phase is observed in the HL-12h sample. Similarly, the PL-6h sample shows the (004) plane of the type II La_2_O_2_CO_3_ phase, which is in good agreement with the XRD and FT-IR data. However, the morphological structures of PL-12h, HL-12h, and PL-6h samples are similar, as shown by the FE-SEM images; the aggregates of nanoparticles have different sizes: smaller than 10 nm for PL-12h, 10–30 nm for HL-12h, and 30–60 nm for PL-6h (Figure 3d–f).

To further understand the formation mechanism of the monoclinic and hexagonal La_2_O_2_CO_3_ phases, uncalcined samples after precipitation or hydrothermal treatment were investigated. The XRD and FT-IR measurements indicate that different chemical products are also produced depending on the preparation conditions (Figure 4). The XRD peaks in Figure 4(Aa–Ac), shown as circles, are indexed to the pure hexagonal phase of La(OH)_3_ with a P63/m(176) space group (JCPDS 36-1481) [1,6,12,21,26,27,28], which clearly shows that the initial La(OH)_3_ phase remains unchanged in the hydrothermal method. With an increase in the preparation time during the hydrothermal method, the crystallinity of the La(OH)_3_ structure becomes stronger with sharper XRD characteristic peaks. Meanwhile, in the precipitation method, the La(OH)_3_ phase is produced with a very low crystallinity for PL-6h (weak and broad characteristic XRD peaks in Figure 4(Ac)). However, when the precipitation time is increased up to 12 h, the characteristic XRD peaks assigned to the orthorhombic La(OH)CO_3_ structure (JCPDS 49-0981) appear with the disappearance of the XRD peaks in the La(OH)_3_ structure (Figure 4(Ad)) [9,29]. Therefore, in the precipitation method, the dominant phase evolves from La(OH)_3_ to La(OH)CO_3,_ with an increase in the precipitation time. However, the initial La(OH)_3_ phase in the hydrothermal method is more crystallized during the hydrothermal treatment.

The FT-IR spectra of the uncalcined samples are monitored to confirm the existence of La(OH)_3_ and La(OH)CO_3_. First, the strong bands at 1438 and 1491 cm^−1^ shown in Figure 4(Bd) can be assigned to the bending vibrations of CO_3_^2−^, which confirms the presence of carbonate species in the intermediate [9]. A band at 3616 cm^−1^ and a broad band at 3410 cm^−1^ represent the O–H stretching mode in La–OH [6,9,27]. The bands at 850 and 1052 cm^−1^ correspond to the vibrational modes of carbon-related bonds, such as CH and CO, which remain before the calcination step. Thus, the FT-IR spectrum in Figure 4(Bd) clearly confirms the existence of La(OH)CO_3_ as an intermediate in the PL-12h sample, which is consistent with the XRD data shown in Figure 4A. For La(OH)_3_, the characteristic FT-IR bands for the O–H stretching and bending modes in La–OH are clearly observed at 3616, 3410, and 1640 cm^−1^, as shown in Figure 4(Ba–Bc) [6,9,27]. Other bands at approximately 2800–3000, 850, and 1052 cm^−1^ can also be assigned to the vibrational modes of carbon-related bonds. Interestingly, for the samples in the precipitation method, the characteristic IR bands for CO_3_^2−^ at approximately 1350–1500 cm^−1^ become sharp and strong with an increase in the reaction time (Figure 4(Bc,Bd)), whereas the characteristic IR band for OH at 3616 cm^−1^ is strongly intensified during the hydrothermal method (Figure 4(Ba,Bb)). Therefore, the precipitation method induces the transformation from La(OH)_3_ into La(OH)CO_3_ through the reaction with CO_2_. In the hydrothermal method, the crystallization of La(OH)_3_ goes further, which results in the high crystallinity of La(OH)_3_.

A critical difference between the two preparations is an open or closed reaction system, which is related to the supply of carbonate sources. For either the precipitation or hydrothermal method, the La precursor in the aqueous solution is dissociated into La cations and is then readily crystallized into the La(OH)_3_ phase, because the initial pH conditions are strongly basic (i.e., pH = 12). In the hydrothermal method, a Teflon-lined autoclave reactor is used as a closed reaction system. Because it is a closed system, there is no further transformation of the La intermediate, which only results in the strong crystallization of the La(OH)_3_ phase for the HL-12h and HL-24h samples. However, in the precipitation method, the precipitation is carried out in an open beaker; thus, the carbonate source (i.e., CO_2_ from the air) can be continuously dissolved into the aqueous solution. Therefore, the initial phase, La(OH)_3_, can be converted into the La(OH)CO_3_ phase by the reaction with CO_2_ at a time longer than 12 h, even though the 6-h precipitation produces only a weakly crystallized La(OH)_3_. Under the continuous CO_2_ supply condition, there is a transformation from La(OH)_3_ into La(OH)CO_3_. In the literature, it was reported that La(OH)_3_ changed into an La carbonate when it was exposed to air [6,27,28]. More importantly, the La(OH)CO_3_ phase is finally converted into the monoclinic type Ia La_2_O_2_CO_3_ phase in the precipitation, while La(OH)_3_ is transformed into the hexagonal type II structure in the hydrothermal method. The sufficient supply of CO_2_ into the aqueous solution produces the La(OH)CO_3_ that can be changed into the monoclinic La_2_O_2_CO_3_ phase.

### 3.2. CO_2_ Behavior on La_2_O_2_CO_3_ Nanoparticles

To investigate the CO_2_ behavior on each La_2_O_2_CO_3_ phase, TGA, CO_2_-TPD and CV of CO_2_ electrochemical reduction for PL-12h (monoclinic type Ia La_2_O_2_CO_3_ phase) and HL-12h (hexagonal type II La_2_O_2_CO_3_ phase) were conducted in this study. Figure 5A shows the derivative TGA (DTGA) profiles of PL-12h and HL-12h, where the decomposition peaks correspond to CO_2_ gases that leave from the La_2_O_2_CO_3_ phases. The weight loss due to the thermal decomposition occurs at 326 °C and in the temperature range of 770–800 °C. According to previous studies [4,30], the CO_2_ peak, which appears during the decomposition of La_2_O_2_CO_3_ above 600 °C, can be assigned to CO_2_ gases leaving from the bulk structure of the La_2_O_2_CO_3_ phases, which is then transformed into the La_2_O_3_ phase. The CO_2_ decomposition from the bulk structure of the hexagonal La_2_O_2_CO_3_ phase occurs at approximately 800 °C, which is higher than the temperature of CO_2_ production during the decomposition of the bulk structure of the monoclinic La_2_O_2_CO_3_ phase. This result shows that the thermal stability of the hexagonal La_2_O_2_CO_3_ phase is higher than that of the monoclinic phase [21]. The weight loss at approximately 326 °C is assumed to be due to the release of CO_2_ gas that is adsorbed on the surface of the La_2_O_2_CO_3_ phase. The decomposition peak at approximately 326 °C has a much smaller intensity than that at 650 °C, which indicates that a much lower amount of CO_2_ is adsorbed onto the surfaces of the La_2_O_2_CO_3_ phase than that released from the bulk structure. Furthermore, based on each peak’s intensity, shown in Figure 5A, the hexagonal type II La_2_O_2_CO_3_ phase contains more CO_2_ on the surface than that on the monoclinic type Ia phase.

To better understand the CO_2_ adsorption ability on the surface of each La_2_O_2_CO_3_ phase, the CO_2_-TPD profiles of PL-12h and HL-12h were acquired. Before conducting the CO_2_-TPD experiments, both samples were thermally treated at 600 °C for 1 h in He gas, and then CO_2_ was introduced into the reactor at 50 °C to perform the CO_2_ adsorption. Therefore, CO_2_ can be assumed to adsorb on the surface of La_2_O_2_CO_3_ phases and then desorb from the adsorption surface sites, which demonstrates the CO_2_ adsorption behavior on the monoclinic and hexagonal La_2_O_2_CO_3_ phases. In Figure 5B, the CO_2_ desorption peaks can be approximately categorized into three types. The peak at approximately 100 °C is related to a weak basic site, and the peaks in the range of 200–400 °C correspond to medium and strong basic sites [2,12,31,32]. The CO_2_ adsorption modes on each basic site have been studied by a combination of FT-IR spectroscopy and CO_2_-TPD measurements [31,32]. Manoilova et al. [30] investigated the CO_2_ adsorption onto La_2_O_3_ by IR spectroscopy, TPD, and DFT calculations. The DFT calculation for the CO_2_ adsorption on La_2_O_3_ predicted that CO_2_ gas adsorbed on the surface in the form of polydentate and monodentate species as a starting structure, and then La_2_O_3_ made a stable connection with polydentate and asymmetric CO_2_ adsorptions at the saturated coverage. The CO_2_ desorption peak at approximately 290 °C in the CO_2_-TPD profile of LaOCl was assigned to the decomposition of coupled bridged CO_2_ adsorbate species [31]. On the basis of the results from the FT-IR and CO_2_-TPD measurements of Mg–Al basic oxides, Di Cosimo et al. [32] suggested that the three types of CO_2_ adsorption modes (e.g., bicarbonate, bidentate carbonate, and unidentate carbonate) were low-strength, medium-strength, and high-strength basic sites, respectively. It was determined that bidentate and unidentate carbonates remained on the surface at approximately 300 °C; only unidentate carbonate was detected at 350 °C [32]. Therefore, in this study, the peak at 110 °C, peaks at approximately 240 °C, and shoulders at approximately 310 °C can be assigned to the desorption of CO_2_ species adsorbed on weak, medium and strong basic sites, respectively. Figure 5B and Table 1 shows that the HL-12h sample has a higher combined intensity of medium and strong basic sites than PL-12h, which suggests that the hexagonal type II La_2_O_2_CO_3_ phase provides more CO_2_ adsorption sites on the surface. This observation is in good agreement with the TGA results shown in Figure 5A.

A DFT calculation was performed to optimize the bulk structures of both La_2_O_2_CO_3_ phases (Figure 6). The lattice constant of La_2_O_2_CO_3_ in the disordered hexagonal structure was predicted by considering the ratio (c/a) of lattice parameters (a and c) of the hexagonal structure [33]. Our DFT calculated lattice constants of La_2_O_2_CO_3_ nanoparticles in both monoclinic and hexagonal structures, similar to the available experimental data from the literature, which are shown in Table 2 [34,35]. On the basis of the DFT calculation, we can optimize the hexagonal type II and monoclinic type Ia La_2_O_2_CO_3_ nanopartilces, as shown in Figure 7. From the optimized structure of each phase, the La atom is determined to have seven and eight oxygen atoms as nearest neighbors in monoclinic and hexagonal structures, respectively. The eight coordination numbers of the La atom in the hexagonal type II La_2_O_2_CO_3_ nanoparticles can produce stronger bonding with carbonate species, which results in the higher stability of the hexagonal type II structure compared to that of the monoclinic type Ia La_2_O_2_CO_3_.

### 3.3. CO_2_ Electrochemical Reduction

Figure 8 shows LSV curves ranging from 0 to −0.6 V vs. Ag/AgCl for PL-12h and HL-12h in CO_2_-saturated 0.1 M NaHCO_3_ electrolyte. HL-12h exhibits a maximum total current density of −25.2 mA/cm^2^ at −1.26 V vs. Ag/AgCl ,whereas a maximum current density of −17.97 mA/cm^2^ for PL-12h is achieved at −1.438 V vs. Ag/AgCl. In addition, HL-12h shows a more positive onset potential toward CO_2_ electrochemical reduction than PL-12h in Figure 8. Both the higher current density and more positive onset potential apparently indicate a higher activity toward the CO_2_ electrochemical reduction in HL-12h compared to that of PL-12h.

The chronoamperometry (CA) experiments were performed at different potentials for each 10 min, and gaseous products were determined by GC. For both PL-12h and HL-12h, the main gaseous products are CH_4_, C_2_H_4,_ C_2_H_6_ and H_2_. Figure 9 shows the Faraday efficiency (FE) of carbon-containing products for PL-12h and HL-12h, resulting in a much higher FE for HL-12h than those for PL-12h. C_2_H_4_ is a dominant carbonaceous product at lower potential. A maximum of the ethene FE (9.4%) for HL-12h is achieved at −0.6 V (vs Ag/AgCl), while that for PL-12h is lower than 5%. Interestingly, CO gas was not detected in the potential range, even for the two La_2_O_2_CO_3_ samples_._ This indicates that La_2_O_2_CO_3_ catalysts are efficient for C-C coupling rather than desorption to form CO gas, since CO is an intermediate for CO_2_ transformation to ethene during CO_2_ reduction [36]. The superior electrocatalytic activity of HL-12h to PL-12h would result from the better CO_2_ adsorption ability which can optimize the first step involving electron and proton transfer to form a *COOH intermediate, which is then converted to other carbonaceous products [37]. The higher electronegativity of hexagonal La_2_O_2_CO_3_ of HL-12h leads to the better CO_2_ adsorption ability [38].

## 4. Conclusions

In this study, La_2_O_2_CO_3_ nanoparticles with hexagonal and monoclinic phases were prepared by different preparation methods, and the CO_2_ behavior on each crystalline structure was investigated by CO_2_-TPD, TGA measurements, and CO_2_ electrochemical reduction. The hydrothermal method produced the hexagonal type II La_2_O_2_CO_3_ phase, whereas the monoclinic type Ia phase was synthesized by the precipitation method (PL-12h and PL-24h). The initial La(OH)_3_ phase was transformed into the La(OH)CO_3_ phase by the reaction with CO_2_ supplied from air in the precipitation method. The hexagonal La_2_O_2_CO_3_ phase showed a higher CO_2_ adsorption ability on the surface and a higher stability in the bulk structure than the monoclinic phase, owing to the differences in optimized crystalline structures predicted by the DFT calculation. Consequently, the hexagonal La_2_O_2_CO_3_ phase of HL-12h had a higher current density and a more positive onset potential than the monoclinic La_2_O_2_CO_3_ of PL-12h in CO_2_ electrochemical reduction.

## Figures and Tables

**Figure 1 nanomaterials-10-02061-f001:**
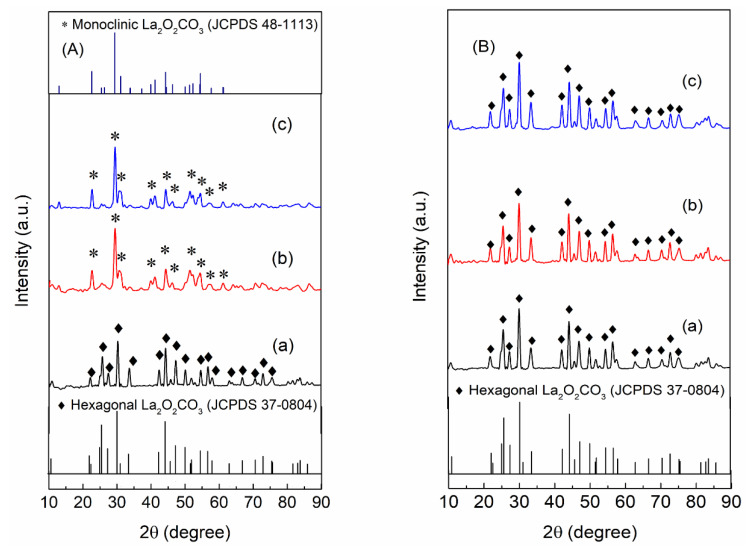
XRD patterns of **A**(**a**–**c**) PL-6h, PL-12h, and PL-24h and **B**(**a**–**c**) HL-6h, HL-12h, and HL-24h.

**Figure 2 nanomaterials-10-02061-f002:**
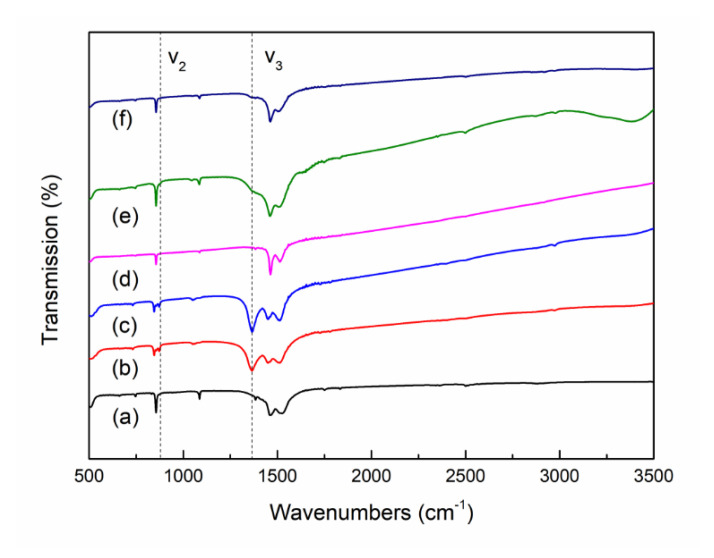
FT-IR spectra of (**a**) PL-6h, (**b**) PL-12h, (**c**) PL-24h, (**d**) HL-6h, (**e**) HL-12h, and (**f**) HL-24h.

**Figure 3 nanomaterials-10-02061-f003:**
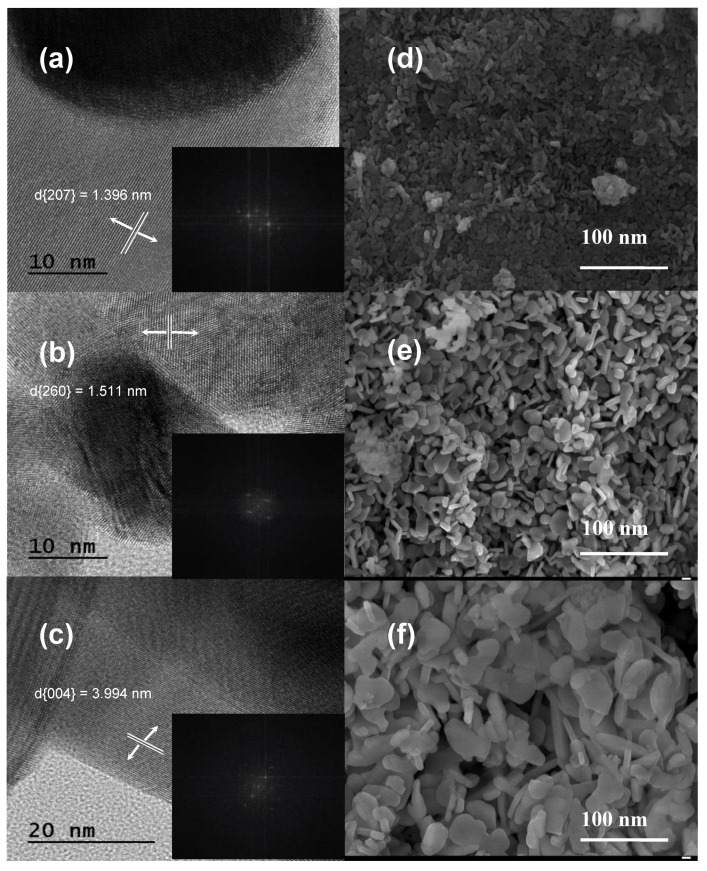
High-resolution transmission electon microscopy (HR-TEM) images of (**a**–**c**) PL-12h, HL-12h, and PL-6h, and FE-SEM images of (**d**–**f**) PL-12h, HL-12h, and PL-6h. The insets of (**a**–**c**) show their fast Fourier transform patterns.

**Figure 4 nanomaterials-10-02061-f004:**
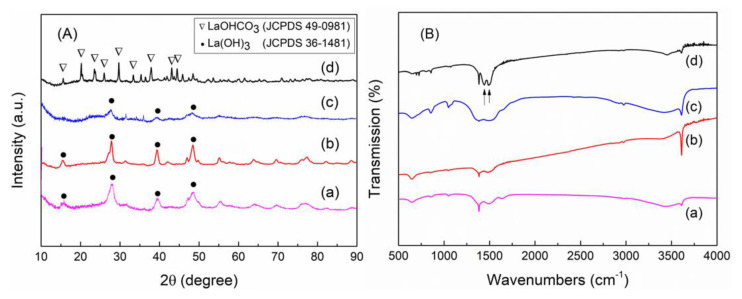
(**A**) XRD patterns and (**B**) FT-IR spectra of uncalcined samples. (**a**) HL-6h, (**b**) HL-12h, (**c**) PL-6h, and (**d**) PL-12h.

**Figure 5 nanomaterials-10-02061-f005:**
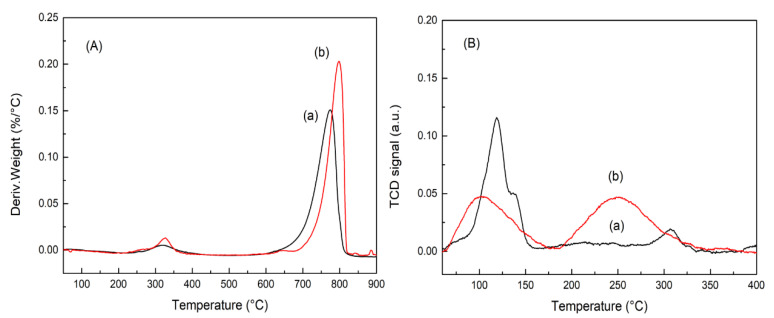
(**A**) Derivative thermogravimetric analysis (TGA) profiles and (**B**) CO_2_-tempurate-programmed desorption (TPD) patterns of (**a**) PL-12h and (**b**) HL-12h.

**Figure 6 nanomaterials-10-02061-f006:**
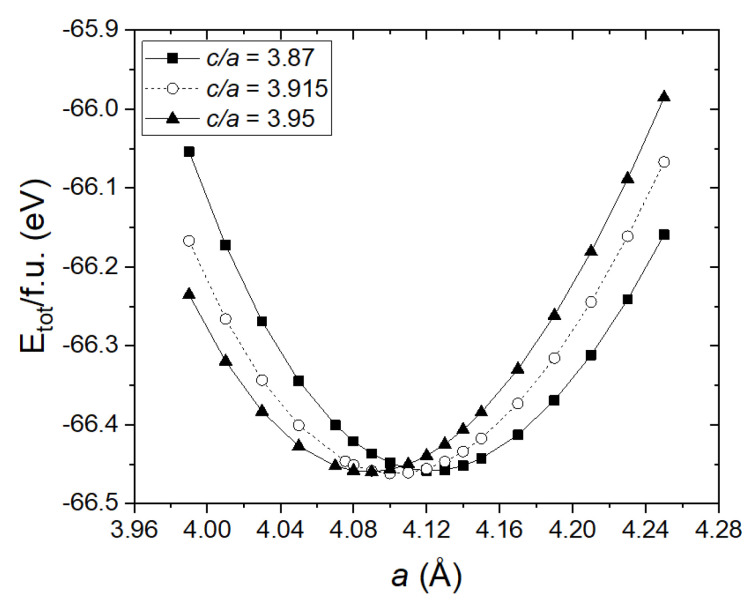
Density functional theory (DFT) total energy (per formula unit) of La_2_O_2_CO_3_ in the hexagonal structure for three c/a values.

**Figure 7 nanomaterials-10-02061-f007:**
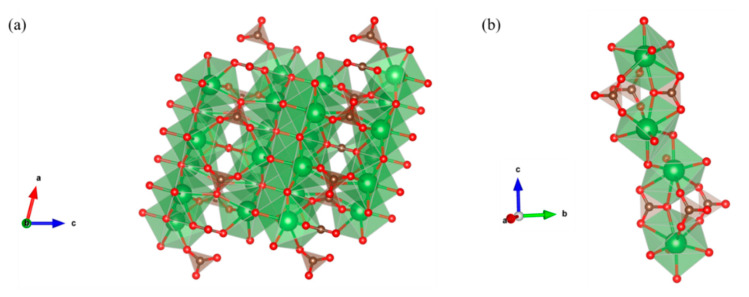
DFT-optimized structure of La_2_O_2_CO_3_ in (**a**) monoclinic and (**b**) hexagonal phases.

**Figure 8 nanomaterials-10-02061-f008:**
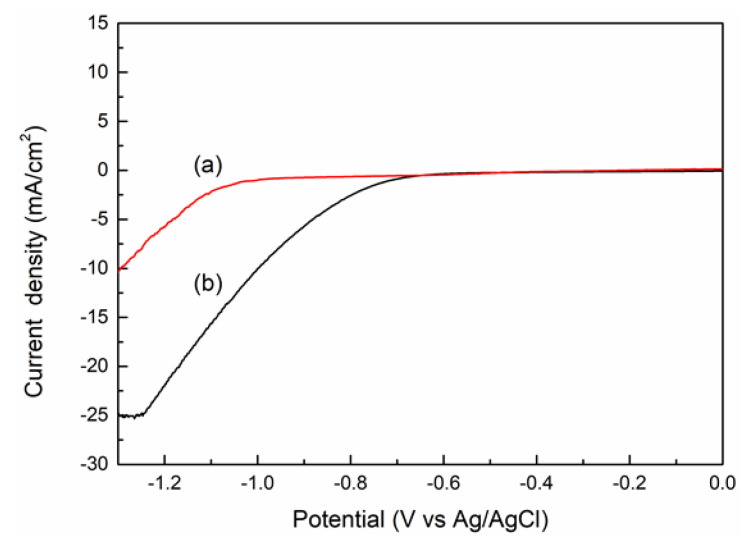
Linear sweeping voltammetry (LSV) curves of electrodes at various reaction times in a 0.1M NaHCO_3_ solution at a scan rate of 20 mV/s: (**a**) PL-12h; (**b**) HL-12h.

**Figure 9 nanomaterials-10-02061-f009:**
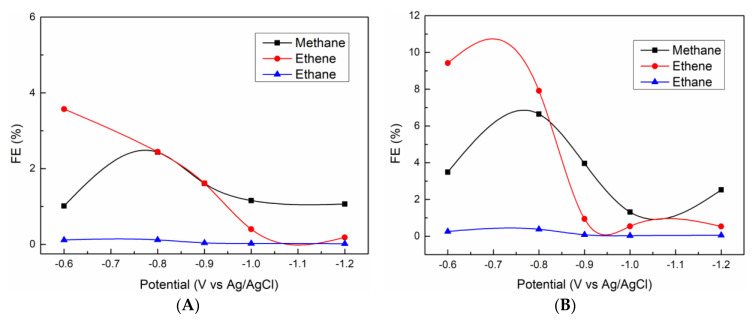
FE values for the (**A**) PL-12h and (**B**) HL-12h as a function of the potential.

**Table 1 nanomaterials-10-02061-t001:** The peak intensities quantified in the CO_2_-TPD patterns.

Samples	Temperature at Maximum (°C)	Quantity (cm^3^/g STP)
PL-12h	119	31.7
	306	3.38
HL-12h	109	24.6
	241	29.0

**Table 2 nanomaterials-10-02061-t002:** Lattice constants of La_2_O_2_CO_3_ in monoclinic and hexagonal structures. The experimental lattice data of monoclinic [34] and hexagonal [35] structures are available from the literature.

La_2_O_2_CO_3_	DFT Calculated Data	Experimental Data [33,34]
Monoclinic	a = 12.286 Å	a = 12.239 Å
	b = 7.097 Å	b = 7.067 Å
	c = 16.531 Å	c = 16.465 Å
	β = 75.677	β = 75.690
Hexagonal	a = 4.100 Å	a = 4.076 Å
	b = 4.100 Å	b = 4.076 Å
	c = 16.053 Å	c = 16.465 Å
	γ = 120	γ = 120

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
