# Peer review of "Hexagonal and Monoclinic Phases of La_2_O_2_CO_3_ Nanoparticles and Their Phase-Related CO_2_ Behavior"

_nanomaterials, 2020, doi:10.3390/nano10102061_

Round 1
Reviewer 1 Report
This is my report as a reviewer for the manuscript entitled "Hexagonal and monoclinic phases of La2O2CO3 nanoparticles and their phase-related CO2 behavior " by Hongyan Yu and co-workers.
The study is very interesting and rich in detailed information. The paper is well written the results have been discussed and the conclusions are supported by the data obtained.
few suggestions and correction,
Abstract
Two lines should be added in which the tools used to characterize the crystalline phases are listed.
Introduction
Pag. 2 line 55-62, FE-SEM-EDX and TEM should be added in Introduction and Abstract sections, as they are used to characterize the phases.
Materials and methods
Pag. 3 line 97, thermogravimetric analyzer (TA …should be uniformed with termogravimetric analysis (TGA) mentioned in Introduction.
linear sweeping voltammetry (LSV) should be added in Introduction and Abstract sections.
Result and discussion
Pag. 3, line 119, P63/mmc space group is hexagonal not monoclinic, please correct it
Fig. 1, please, enlarge the symbols and writings in the figure.
Pag. 4, line 143-151, I believe it would be better to integrate the TEM figures into the text, and not as supplementary material.
Fig. 3, please, enlarge the symbols and writings in the figure.
Please, the acronym DTGA must be made explicit, in Materials and methods section.
Caption of Figure 5, the numbers of the formula go in subscript
Conclusion
Line 308, …”with long preparation time (12 h and 24 h)” wasn't it the same time for both types of synthesis?
Apart from this minor quibble, the paper looks to be in good shape for publication.
Author Response
Reviewer#1
Abstract
Two lines should be added in which the tools used to characterize the crystalline phases are listed.
Response) We appreciate this suggestion. We have modified the sentence in the revised manuscript as follows:
“In this study, we prepared hexagonal and monoclinic phases of La2O2CO3 nanoparticles by different wet preparation methods and investigated their phase-related CO2 behavior through field-emission scanning microscopy, high-resolution transmission electron microscopy, Fourier transform infrared, thermogravimetric analysis, CO2-temperature programmed desorption, and linear sweeping voltammetry of CO2 electrochemical reduction.” (lines 12-15 on the revised manuscript)
Introduction
Pag. 2 line 55-62, FE-SEM-EDX and TEM should be added in Introduction and Abstract sections, as they are used to characterize the phases.
Response) We are thankful for the useful suggestion. We have modified the sentences in the revised manuscript as follows:
“In this study, we prepared the nanoparticles with type Ia and type II La2O2CO3 crystal structures by conventional wet preparation methods and investigated the formation of different La2O2CO3 phases with Fourier transform infrared (FT-IR), X-ray diffraction (XRD), field-emission scanning electron microscopy (FE-SEM), and high-resolution transmission electron microscopy (HR-TEM).” (lines 60-62 on the revised manuscript)
“The morphologies of the samples were observed by a field-emission scanning electron microscope (JEOL, JSM-600F, Tokyo, Japan) instrument equipped with an energy dispersive spectrometer. HR-TEM images were obtained using a JEOL JEM-2100F instrument (JEOL Ltd., Tokyo, Japan). (lines 86-89 on the revised manuscript)
Materials and methods
Pag. 3 line 97, thermogravimetric analyzer (TA …should be uniformed with termogravimetric analysis (TGA) mentioned in Introduction.
Response) We appreciate this suggestion. However, in this part, we would like to introduce an instrument used for the characterization method. Therefore, we need to keep the former sentence.
linear sweeping voltammetry (LSV) should be added in Introduction and Abstract sections.
Response) We are thankful for the reasonable suggestion. We have modified the sentences in the revised manuscript as follows:
“In this study, we prepared hexagonal and monoclinic phases of La2O2CO3 nanoparticles by different wet preparation methods and investigated their phase-related CO2 behavior through field-emission scanning microscopy, high-resolution transmission electron microscopy, Fourier transform infrared, thermogravimetric analysis, CO2-temperature programmed desorption, and linear sweeping voltammetry of CO2 electrochemical reduction.” (lines 12-15 on the revised manuscript)
“Furthermore, the CO2 behavior on the different La2O2CO3 crystal structures was observed by CO2-temperature programmed desorption (TPD), thermogravimetric analysis (TGA), and linear sweeping voltammetry (LSV) of CO2 electrochemical reduction.” (lines 62-64 on the revised manuscript)
“The CO2 electrochemical reduction was carried out via the LSV measurement with an Ag/AgCl electrode as a reference electrode and Pt wire as a counter electrode.” (lines 104-106 on the revised manuscript)
Result and discussion
Pag. 3, line 119, P63/mmc space group is hexagonal not monoclinic, please correct it
Response) We are grateful for the useful comment. We have modified it in the revised manuscript as follows:
“However, when the precipitation time is increased to 12 h and 24 h, the characteristic XRD peaks of the monoclinic La2O2CO3 phase clearly appear at 2θ = 22.8, 29.3, 31.0, 39.9, and 44.4° with a C12/c1 space group (JCPDS 48-1113)” (lines 125-127 on the revised manuscript)
Fig. 1, please, enlarge the symbols and writings in the figure.
Response) As requested by the reviewer, we have modified Fig. 1.
Pag. 4, line 143-151, I believe it would be better to integrate the TEM figures into the text, and not as supplementary material.
Response) As requested by the reviewer, we have added the images (Figure 3 in the revised manuscript) into the text. The corresponding figure numbering has been modified in the revised manuscript, too.
Fig. 3, please, enlarge the symbols and writings in the figure.
Response) As requested by the reviewer, we have modified Fig. 4 in the revised manuscript.
Please, the acronym DTGA must be made explicit, in Materials and methods section.
Response) We appreciate this comment. However, we have modified the caption of Figure 5 in the revised manuscript as follows:
“Figure 5. (A) Derivative TGA profiles and (B) CO2-TPD patterns of (a) PL-12h and (b) HL-12h.” (line 237 on the revised manuscript)
Caption of Figure 5, the numbers of the formula go in subscript
Response) As requested by the reviewer, we have modified it in the revised manuscript.
“Figure 6. DFT total energy (per formula unit) of La2O2CO3 in the hexagonal structure for three c/a values.” (lines 280-281 on the revised manuscript)
Conclusion
Line 308, …”with long preparation time (12 h and 24 h)” wasn't it the same time for both types of synthesis?
Response) We appreciate this comment. We have changed the sentence in the revised manuscript as follows:
“The hydrothermal method produced the hexagonal type II La2O2CO3 phase, whereas the monoclinic type Ia phase was synthesized by the precipitation method (PL-12h and PL-24h).” (lines 318-320 on the revised manuscript)
Apart from this minor quibble, the paper looks to be in good shape for publication.
Response) We greatly appreciate this comment.
Reviewer 2 Report
The paper “Hexagonal and monoclinic phases of La2O2CO3 nanoparticles and their phase-related CO2 behavior” by H. Yu et al (nanomaterials-964236) describes the synthesis of hexagonal and monoclinic phases of lanthanum oxycarbonate (La2O2CO3) nanoparticles by two methods (a conventional precipitation and a hydrothermal method). Their phase-related CO2 behavior is also studied by different techniques. The results are also analyzed in the light of DFT calculations.
The study is well planned, and the results are consistent with the conclusions. However, I consider that the authors should adress the suggestions I make and answer the questions that follow; so my decision is that the article may be acceptable for publication in Nanomaterials after minor revision.
Both the Abstract and the Introduction are well conceived except for the paragraph between lines 60-62, where the authors should limit themselves to describing the study without drawing conclusions.
In point 2.2. Characterizations: a) The temperature corresponding to the X-ray diffraction study data collection must be specified. b) In line 96, letter “i” is missing in “increased”. c) In lines 109-110, the authors should explain the values related to the convergence criteria.
In 3.1. Synthesis of monoclinic and hexagonal La2O2CO3 nanoparticles: d) In the sentence corresponding to lines 129-31, a bibliographic reference regarding the assignments of FT-IR bands for carbonates in La2O2CO3 must be included. e) The authors refer to images (Figures S1(a-c)) obtained by electron microscopy (HR-TEM and FE-SEM) of the samples at different times (lines 149-151). In my opinion, that Figure can be included in the manuscript itself and thus eliminate Supplementary Materials. On the other hand, can the authors provide any reasoning to explain why the morphology is larger in PL-6h?
In Section 3.2. CO2 behavior on La2O2CO3 nanoparticles: f) After establishing that the peaks at approximately 100oC correspond to the desorption of CO2 species adsorbed on weak basic sites and that those at 240 and 310oC are indicative of medium and strong sites, they refer to Figure 4B to conclude that “…the HL-12h sample has higher intensities of medium and strong basic sites than those of PL-12h…”. However, in view of that Figure, it is observed that it is the weak and strong sites that show the highest signal in the CO2-TPD patterns for the HL-12h sample, the same as Table 1 indicates. g) In lines 260-262, the authors state “Our DFT calculated lattice constants of La2O2CO3 nanoparticles in both monoclinic and hexagonal structures are similar to the available experimental data from the literature, which are shown in Fig. 5 and Table 2”. This paragraph is not clear: has Fig. 5 been taken from literature? The authors should clarify these last two points (f, g).

Author Response
Reviewer #2
Both the Abstract and the Introduction are well conceived except for the paragraph between lines 60-62, where the authors should limit themselves to describing the study without drawing conclusions.
Response) As requested by the reviewer, we have modified the sentence as follows:
“The superior CO2 behavior of the hexagonal La2O2CO3 phase to the monoclinic phase was additionally explained by the crystalline structures of both La2O2CO3 phases, which was predicted by the density functional theory (DFT) calculation.” (lines 64-67 on the revised manuscript)
In point 2.2. Characterizations:
a) The temperature corresponding to the X-ray diffraction study data collection must be specified.
Response) We are thankful for this comment. We have modified it in the text as follows:
“XRD patterns were measured at room temperature on a Rigaku D/MAX-2200 powder X-ray diffractometer” (lines 90-91 on the revised manuscript)
b) In line 96, letter “i” is missing in “increased”.
Response) We are thankful for this comment. We have modified it in the revised manuscript.
c) In lines 109-110, the authors should explain the values related to the convergence criteria.
Response) As requested by the reviewer, we have modified the text in the revised manuscript as follows:
“The criteria of convergence of energies and forces for geometry optimization were 10−4 eV and 10−2 eV/Å, respectively. For the calculation of disordered hexagonal La2O2CO3, the lowest energy configuration among the other randomly selected 50 structures was used. The Monkhorst-Pack k-point meshes of 3 × 5 × 2 and 9 × 9 × 3 were used for the geometry optimization of monoclinic and hexagonal phase of La2O2CO3, respectively [25].” (lines 115-118 on the revised manuscript)
In 3.1. Synthesis of monoclinic and hexagonal La2O2CO3 nanoparticles:
d) In the sentence corresponding to lines 129-31, a bibliographic reference regarding the assignments of FT-IR bands for carbonates in La2O2CO3 must be included.
Response) The references were cited after the next sentence. However, we have moved the citation into the sentence that the reviewer mentioned in the revised manuscript as follows:
“According to the assignments of typical FT-IR bands for carbonates in the La2O2CO3 phases, the bands at 745, 855, 1066, and 1518 cm−1 are interpreted as CO32− vibrations related to the La2O2CO3 phase [4,6,21,27].” (lines 135-137 on the revised manuscript)
e) The authors refer to images (Figures S1(a-c)) obtained by electron microscopy (HR-TEM and FE-SEM) of the samples at different times (lines 149-151). In my opinion, that Figure can be included in the manuscript itself and thus eliminate Supplementary Materials. On the other hand, can the authors provide any reasoning to explain why the morphology is larger in PL-6h?
Response) As requested by the reviewer, we have inserted the images (Figure 3 in the revised manuscript) into the main text. The corresponding figure numbering has been modified in the revised manuscript, too. However, we have no idea on the large nanoparticle size of PL-6h. We are very sorry not to explain it.
In Section 3.2. CO2 behavior on La2O2CO3 nanoparticles:
f) After establishing that the peaks at approximately 100oC correspond to the desorption of CO2 species adsorbed on weak basic sites and that those at 240 and 310oC are indicative of medium and strong sites, they refer to Figure 4B to conclude that “…the HL-12h sample has higher intensities of medium and strong basic sites than those of PL-12h…”. However, in view of that Figure, it is observed that it is the weak and strong sites that show the highest signal in the CO2-TPD patterns for the HL-12h sample, the same as Table 1 indicates.
Response) In the text, even though we assign three CO2 adsorption species on the materials, we can quantify two CO2 desorption peaks of different temperature at maximum, shown in Table 1. The peak around 110 ℃ was assigned to the CO2 adsorbed on weak basic sites. Both the materials have large quantity of the weak CO2 adsorption site (31.7 cm3/g STP for PL-12h and 24.6 cm3/g STP for HL-12h). However, for the medium and strong CO2 adsorption sites, it was hard to deconvolute them in the TPD patterns and we calculated the combined quantities of CO2 adsorbed on the medium and strong basic sites in Table 1. In this aspect, it is clear that HL-12h has higher intensity for medium and strong basic sites than those of PL-12. We have modified the sentence in the revised manuscript as follows:
“Figure 5B and Table 1 shows that the HL-12h sample has higher combined intensity of medium and strong basic sites than PL-12h” (lines 261-262 on the revised manuscript)
g) In lines 260-262, the authors state “Our DFT calculated lattice constants of La2O2CO3nanoparticles in both monoclinic and hexagonal structures are similar to the available experimental data from the literature, which are shown in 5 and Table 2”. This paragraph is not clear: has Fig. 5been taken from literature? The authors should clarify these last two points (f, g).
Response) We are sorry to make this confusion. Table 2 shows the similarity between DFT calculation results in this study and the experimental data. Therefore, we have deleted Fig. 5 from the sentence in the revised manuscript as follows:
“Our DFT calculated lattice constants of La2O2CO3 nanoparticles in both monoclinic and hexagonal structures are similar to the available experimental data from the literature, which are shown in Table 2 [34,35]” (lines 269-271 on the revised manuscript)